# Classification of Firewall Log Data Using Multiclass Machine Learning Models

Malak Aljabri [1,2,*], Amal A. Alahmadi [3], Rami Mustafa A. Mohammad [4], Menna Aboulnour [2], Dorieh M. Alomari [5] and Sultan H. Almotiri [1]

1 Department of Computer Science, College of Computer and Information Systems, Umm Al-Qura University, Makkah 21955, Saudi Arabia; shmotiri@uqu.edu.sa
2 SAUDI ARAMCO Cybersecurity Chair, Department of Computer Science, College of Computer Science and Information Technology, Imam Abdulrahman Bin Faisal University, P.O. Box 1982, Dammam 31441, Saudi Arabia; 2180007190@iau.edu.sa
3 Department of Networks and Communications, College of Computer Science and Information Technology, Imam Abdulrahman Bin Faisal University, P.O. Box 1982, Dammam 31441, Saudi Arabia; aaalahmadi@iau.edu.sa
4 Department of Computer Information Systems, College of Computer Science and Information Technology, Imam Abdulrahman Bin Faisal University, P.O. Box 1982, Dammam 31441, Saudi Arabia; rmmohammad@iau.edu.sa
5 SAUDI ARAMCO Cybersecurity Chair, Department of Computer Engineering, College of Computer Science and Information Technology, Imam Abdulrahman Bin Faisal University, P.O. Box 1982, Dammam 31441, Saudi Arabia; 2180007089@iau.edu.sa
* Correspondence: msaljabri@iau.edu.sa or mssjabri@uqu.edu.sa

**Abstract:** These days, we are witnessing unprecedented challenges to network security. This indeed confirms that network security has become increasingly important. Firewall logs are important sources of evidence, but they are still difficult to analyze. Artificial Intelligence (AI), Machine Learning (ML), and Deep Learning (DL) have emerged as effective in developing robust security measures due to the fact that they have the capability to deal with complex cyberattacks in a timely manner. This work aims to tackle the difficulty of analyzing firewall logs using ML and DL by building multiclass ML and DL models that can analyze firewall logs and classify the actions to be taken in response to received sessions as "Allow", "Drop", "Deny", or "Reset-both". Two sets of empirical evaluations were conducted in order to assess the performance of the produced models. Different features set were used in each set of the empirical evaluation. Further, two extra features, namely, application and category, were proposed to enhance the performance of the proposed models. Several ML and DL algorithms were used for the evaluation purposes, namely, K-Nearest Neighbor (KNN), Naïve Bayas (NB), J48, Random Forest (RF) and Artificial Neural Network (ANN). One interesting reading in the experimental results is that the RF produced the highest accuracy of 99.11% and 99.64% in the first and the second experiments respectively. Yet, all other algorithms have also produced high accuracy rates which confirm that the proposed features played a significant role in improving the firewall classification rate.

**Keywords:** machine learning; deep learning; network security; firewalls; random forest

## 1. Introduction

The internet is continuously growing exponentially. This substantial growth has given rise to a risk caused by cyber threats. Over the past decade, a significant increase in cyber threats in companies of all industries has been observed. Cyberthreats such as ransomware, phishing, data leakage, hacking, and insider threat put organizations in a risky position. This signifies the necessity to apply mechanisms in order to protect data integrity and usability [1]. In most cyberattacks, the attacker is aware of the defense mechanisms applied by the organization and thus is able to hide his attack. In order to detect such attacks, all

traffic records need to be analyzed continuously to create a profile that aids in determining rules for the firewall to take the appropriate action in response to the received packets. However, those rules are constantly changing to adapt to the variations in attacks, the advancement of the tools, and the sophistication in the attackers' methods [2]. This constant change in the rules is a significant issue as the rules are set manually by the organizations' engineers and security personnel based on their policies and requirements.

Furthermore, firewalls essentially act as control gates for network packets to pass through. The system administrators set up firewalls according to their organization's requirements [3]. As a result of their significant role in securing the network against any internal or external threats, firewalls have proven to be a crucial part of today's communication networks. Fundamentally, firewalls arrange network log records according to their rules which are set manually or by default based on certain criteria, such as the reason for the connection, which ports are communicational, which subdivisions are authorized, etc. These rules may vary as they depend on the organization utilizing the firewall. Additionally, due to advancements and the constantly changing behavior of the environment, updating these rules is a demanding and continuous process [4]. Based on these rules and many other attributes of the network log records, actions are taken, namely, "Allow", "Drop", "Deny", or "Reset-both". Choosing an incorrect action to handle a session may lead to security vulnerabilities, thus allowing undesirable events such as devices' shutdown, loss of service, indirectly causing profit loss, or the release of confidential files.

Hence, firewalls are important elements of any network's security. However, managing and setting the rules that determine the appropriate actions has become complicated and error-prone [4]. Artificial Intelligence (AI) techniques, such as Machine Learning (ML) and Deep Learning (DL) have been proving immense potential in many fields, including cybersecurity [5]. In the field of cybersecurity, there is an increasing awareness of the beneficial application of AI as it can improve defense and network security measures, making systems more robust, resilient, and responsive. Furthermore, network security systems, such as Intrusion Detection Systems (IDS) and firewalls, produce large amounts of log files. Within these records, valuable knowledge is concealed. Using ML, this knowledge, along with the patterns in the network traffic attributes, can be discovered and exploited to produce models that support the detection of network threats [6]. Hence, the necessity to apply and utilize ML and DL algorithms is increasing, to aid and automate the process of predicting the recommended action and comparing it with the applied action [7,8]. Training these systems can generate alerts when threats are detected, identify new types of malware, and protect confidential information for organizations [9]. Moreover, ML and DL techniques have been indispensable tools for years due to their ability to make better decisions without human intervention to facilitate efficient analysis on a larger scale.

In this paper, we aim to use ML classification and clustering algorithms to classify the recommended action as "Allow", "Drop", "Deny", or "Reset-both" [2]. We conducted a comparative performance study for a set of ML and DL algorithms, namely, K-Nearest Neighbor (KNN), Naïve Bayas (NB), J48, Random Forest (RF), and Artificial Neural Network (ANN), which were applied to a dataset collected from the network log of a private organization. We conducted two experiments, each with different sets of features.

The main contributions of this paper are:

- Studying the correlation of the features related to firewall logs of private networks to identify the significance of each feature and adding two extra features to existing literature that improve model performance.
- Conducting a comparative performance evaluation for a set of multiclass ML and DL models to determine the best algorithm to decide the actions to be taken.
- Implementing the experiments using a large-scale real-world dataset that was collected for this work.

This paper is organized as follows: Section 2 summarizes the related work carried out by previous researchers. Section 3 demonstrates the research methodology including the dataset description, pre-processing, feature extraction and selection, performance measure-

ments, and the applied ML and DL algorithms. Section 4 demonstrates the experimental results. Section 5 includes a discussion of the achieved results and Section 6 concludes the paper.

## 2. Literature Review

Many studies were conducted to analyze network logs for various purposes. Several studies aimed to detect the attacks in the records. Other studies focused on identifying the anomalies in the rules rather than in the records, while a few studies matched our goal of identifying the action to be taken to handle received traffic.

Binary classification of a network log was carried out by multiple studies. These papers worked on analyzing network traffic logs and classifying them as normal traffic or anomalous traffic. Allagi et al. [10] followed this approach to identify anomalies in access patterns using supervised ML techniques. The dataset used was taken from a publicly available dataset in the UCI ML repository [11], which consists of 22,614,256 records. Moreover, the Self-Organizing Feature Map (SOFM) algorithm and K-means were applied to train the models. The model was tested using the sample dataset and obtained an accuracy of 97.2% and a False Positive Rate (FPR) of 2.7%. Similarly, Cao et al. [12] proposed enhanced traditional network log analysis mechanisms by developing an Anomaly Detection System (ADS) which is a two-level ML algorithm. The study was performed on data collected from a network log assessment project from an IT security company consisting of 8000 records. Furthermore, six features were extracted based on the knowledge and experience of the security engineers regarding the necessary attribute to consider in order to detect anomalies. At the first level, a binary classifier was used to discriminate between normal and anomalous records. As for the second level, it used the Hidden Markov Model (HMM) to identify and specify the anomaly type. To explore the contribution of the ADS, its performance was compared against three single-level anomaly detection methods based on ML algorithms, which are Support Vector Machines (SVM), Decision Tree (DT), and Logistic Regression (LR). Compared with these single-level ML algorithms, the ADS system obtained better results with a classification accuracy of 93.54% and an FPR of 4.09%. Likewise, As-Suhbani et al. [13] proposed a meta classifier model using four binary classifiers. The network log dataset, consisting of 5,000,000 records, was generated from Snort, an open-source Intrusion Prevention System (IPS) based on rule matching, and Taut Wire Intrusion Detection System (TWIDS), a freeware for endpoint protection was analyzed, and the six extracted features were inserted into ML classifiers, including KNN, NB, J48, and One R using Spark in the Weka tool. The action attribute which takes values of "Allow" or "Drop" was the class attribute. The performance of the four algorithms was compared and evaluated in terms of accuracy, F-measure, and Receiver Operating Characteristics (ROC) values. It was observed that the highest accuracy of 99.87% was achieved by the KNN classifier.

Furthermore, Jia et al. [14] implemented a network log analysis using data mining and ML approaches by combining ML, data mining, and statistical learning in their work. They applied a filtering approach prior to processing and implemented a spark-based log analyzer that was built to enable detect abnormal network behavior through analyzing large-scale log data. The system has advantages in terms of accuracy, timeliness, and scalability in network anomaly detection. However, the attack detection model is not dynamically generated as the records require manual manipulation to be suitable for use by the models. In the same manner, another study was performed by Winding et al. [6], in which data mining and ML were combined and applied to discover network traffic anomalies in network logs. This study aimed to observe the firewall with ML methods and the JRip algorithm to determine if threats could be identified based on the statistical analysis of the logs. Furthermore, nine features were extracted and used to derive four other features. The results of these experiments achieved an accuracy of 99.9167%. However, more research and analysis on feature extraction can lead to better results. For instance, as mentioned in the paper, correlating features obtained from the applied IDS with other features derived from the network log could result in a richer feature vector. Furthermore,

research conducted by Schindler [2] aimed to demonstrate a practical and quick method to analyze real-world network logs to detect breach attempts. To achieve this goal, they proposed to use a modified kill chain model as an indicator of compromise. In this paper, they developed multi-class SVMs and one-class SVM models which achieved an accuracy of 95.33% and 98.67%, respectively. In addition, it was revealed that the abstracted event sequence graph model for mapping to a kill chain enhances the automated forensic analysis of network logs.

Other studies employed binary classification, but their focus was to identify rule anomalies. Ucar et al. [4], proposed an ML-based model to detect anomalies in the firewall rules repository by analyzing a file containing approximately 5,000,000 entries. First, the network logs were analyzed and 17 features were extracted and inserted into a set of ML classification algorithms including HyperPipes, NB, KNN, and DT to determine if an anomaly was found or not. The performance of the algorithms was evaluated using the F-measure. In their experiment, the KNN algorithm demonstrated the best performance with an accuracy of 100%.

In addition, the following three papers were more closely related to our study aiming to identify the action to be taken. Those papers used a part of log records collected from a firewall device used at Firat University [11] containing 65,532 instances. In addition, through feature extraction, 11 features were obtained and used by all of the following three papers. The main paper that used and initially collected the dataset [11] was formulated by Ertam et al. [3]. In the study, the network log was classified using the SVM algorithm. The model's performance was evaluated each time using a different SVM activation function. They performed a multiclass classification of the action attribute which takes the values "allow", "drop", "deny", or "reset-both". The highest recall value of 98.5% was achieved when the sigmoid activation function was used. The best precision of 67.5% was achieved when the linear activation function was used. Furthermore, the highest F-measure of 76.4% was obtained when the Radial Basis Function (RBF) activation function was used. Similarly, a DT classification algorithm was developed by AL-Behadili [15] for network log analysis to predict the action. The same 11 features were selected, and six benchmark classification algorithms were used, which are SVM, One R, ANN, Multiclass classifier, Particle Swarm Optimization (PSO), and ZeroR. The best accuracy of 99.839% was obtained using the DT model. Likewise, Sharma et al. [16] used the same dataset and set of 11 features to train their models. In this study, they used five algorithms, LR, KNN, DT, SVM, and stochastic gradient descent classifier. The highest performance was achieved by the DT classifier which achieved a precision of 87%. However, when they used a stacking ensemble with RF as its meta, they achieved a precision of 91% and an accuracy of 99.8%.

We sought to surpass previous efforts made by similar studies by improving the work they produced. All three papers [4,14,15] aimed to build the optimal model for the analysis of network logs. In our study, we used common algorithms that were applied in similar papers. Moreover, we had a dataset privately collected for our study, 112,532 instances of it were used to train our models. Thus, the number of instances we used is almost double the number of instances used in all three papers which used the UCI repository collected from Firat University [11]. Optimally, this will aid in developing more reliable models and more promising results. Additionally, we conducted experiments with different sets of features to identify their significance as one experiment included 11 features and the second experiment included 2 extra features to include a total of 13 features.

Table 1 summarizes all the previous efforts discussed, including the goal of the paper, the dataset used, the extracted features, the models applied, and the best performance achieved.

**Table 1.** Summary of the literature review.

| Authors | Year | Main Goal | Dataset | Number of Features | Applied Models | Results |
|---|---|---|---|---|---|---|
| Allagi et al. [10] | 2019 | Develop binary classifiers to distinguish between normal and analogous records. | Public network log from UCI ML repository. 22,614,256 instances. | - | Supervised ML approach. K-means and SOFM algorithms. | Accuracy: 97.2% FPR: 2.7%. |
| Cao et al. [12] | 2017 | Build a system for anomaly detection in network log files. | Private security company network log. 8000 instances. | 6 features. | First level: SVM, LR, or DT. Second level: HMM | Accuracy: 93.54% FPR: 4.09%. |
| As-Suhbani et al. [13] | 2019 | Use ML classifiers to analyze network log datasets. | Private network log from their department. 500,000 instances. | 6 features. | NB, KNN, One R, and J48. | Accuracy: (KNN classifier) 99.87% |
| Jia et al. [14] | 2017 | Introduce a Spark-based data security platform to detect abnormal network behavior. | Private multi-source heterogeneous network log data. | - | Data mining, ML, and statistical analysis technologies. | The system has advantages regarding timeliness, accuracy, and scalability in network anomaly detection. |
| Winding et al. [6] | 2006 | Determine if statistical analysis of network logs is suitable to detect threats. | Private production university data center network log. 2401 instances. | - | ML methods and JRip algorithm. | Accuracy: 99.9167% |
| Schindler [2] | 2017 | Perform forensic analysis of network log data and detect attack patterns. | Public KDD-Cup 99/DARPA 1999 datasets. 1228 instances. | - | Multi-class and one-class SVMs. | SVM: 95.33% One-class SVMs: 98.67% |
| Ucar et al. [4] | 2017 | Analyze network log files to discover firewall rules anomalies. | Private data extracted from a firewall. | 17 features. | NB, KNN, DT and HyperPipes. | Accuracy: (KNN classifier) 100% |
| Ertam et al. [3] | 2018 | Compare three multiclass SVM activation functions in determining the action. | Public network log from Firat University. 65,532 instances. | 11 features. | SVM with Linear, polynomial, sigmoid, and RBF activation functions. | Recall: (SVM with the Sigmoid activation function) 98.5%. Precision: (SVM with the linear activation function) 67.5%. F-measure: (SVM with the RBF activation function) 76.4%. |
| AL-Behadili [15] | 2021 | Use multiclass ML to predict the action. | Public network log from Firat University. 65,532 instances. | 11 features. | DT, SVM, One R, ANN, PSO, and ZeroR. | Accuracy: (DT) 99.839% |
| Sharma et al. [16] | 2021 | Build multiclass ML for the classification of network logs. | Public network log from Firat University. 65,532 instances. | 11 features. | KNN, LR, SVM, DT, and stochastic gradient descent classifier. Plus, stacking ensemble with RF as its meta. | Precision: (DT) 87% Precision (Stacking Ensemble): 91% Accuracy: (Stacking Ensemble) 99.8% |

### 3. Materials and Methods

The main aim of this study is to use ML and DL techniques to analyze network traffic datasets and show the impact of adding two extra features to the set of features commonly used in related studies. The utilized models include RF, J48, NB, KNN, and ANN. The models aim to predict the recommended action to be taken with respect to each session as traffic flows through the network. Furthermore, we assessed the performance of these classifiers in terms of accuracy, precision, recall, F-measure, and ROC. We used 10-fold class validation to evaluate the models. The 10-fold validation technique firstly divides the dataset into ten equal sub-datasets where one of them is used for training the model and the remaining nine sub-datasets are used for validation. This process is repeated ten times. Therefore, each sub-dataset is used one time for training and nine times for validation. At the end of each cycle, the evaluation metrics are calculated and after the end of the ten cycles the average of each evaluation metric is calculated and produced as final results. Furthermore, we performed two experiments using different sets of features and tested the significance of each feature. Figure 1 summarizes the steps of the conducted methodology.

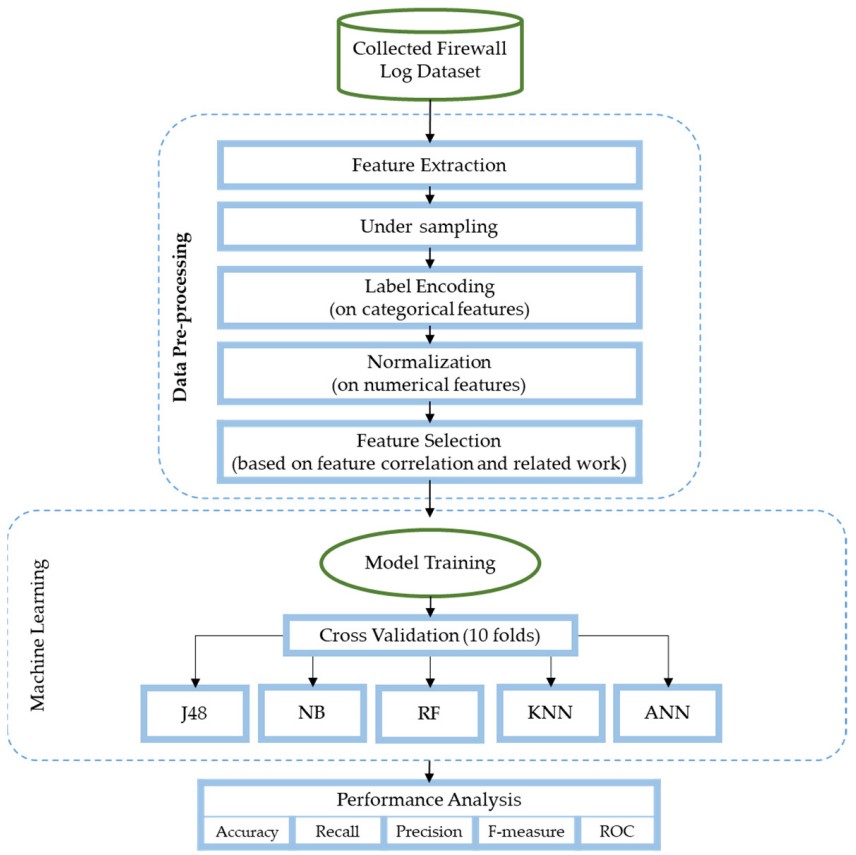

**Figure 1.** Research methodology steps.

### 3.1. Dataset Description

In our study, we were provided with a network log dataset collected by a private organization. The network traffic logs were collected from the 18th to the 27th of May 2021. The log records used were taken from a firewall employed by the organization. The CSV file originally contained 1,048,576 instances. Table 2 shows the statistics of the dataset records.

**Table 2.** Original dataset statistics.

| Feature | Action | |
|---|---|---|
| | Allow | 925,151 |
| Values | Deny | 28,133 |
| | Drop | 42,018 |
| | Reset-both | 53,183 |

### 3.2. Data Preprocessing

Naturally, pre-processing is applied to the dataset before any analysis to make the file ready for use by the models for training and testing. Pre-processing involves loading the dataset, cleaning, manipulating, and converting the data into a form that is suitable for the desired purpose.

The log dataset contained more than one million entries, only 2.68% of which were sessions in which the action taken was "Deny", while sessions for which the action that was taken was "Allow" represented 88.23% of the data. This shows that the data suffered from imbalance where at least one of the class labels was not balanced in number when compared to other class labels.

Considering the dataset was highly unbalanced, under-sampling was applied to equalize the number of instances of each action type, as demonstrated in Figure 2. Hence, 28,133 instances of each action type were randomly selected to have a total of 112,532 instances. As for the categorical features, label encoding was then used to convert them to numerical values. Moreover, the numerical features were normalized to within a range of −1:1.

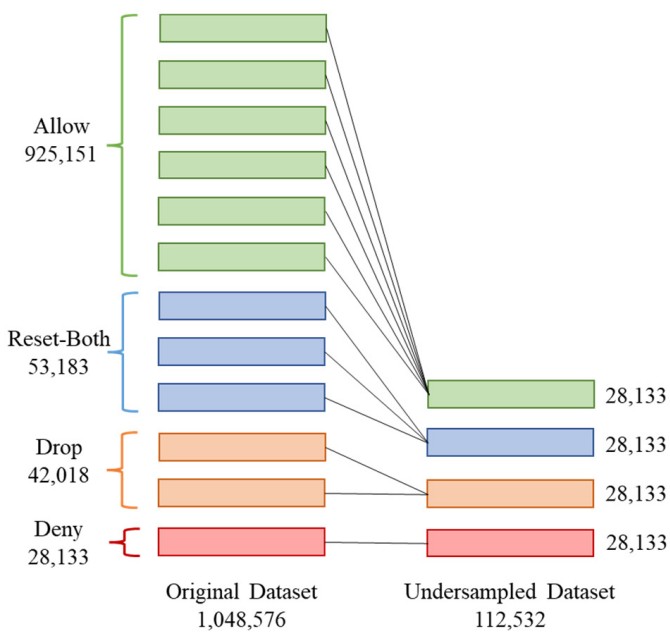

**Figure 2.** Dataset before and after randomized under-sampling was applied.

### 3.3. Feature Selection

The dataset included 25 categorical and numerical features describing the network traffic sessions, the target feature, and the action. We conducted two different experiments to obtain the highest possible performance. A list of the features used in each experiment is listed in Table 3. The action attribute was selected as the class attribute which takes four values of "Allow", "Drop", "Deny", or "Reset-both".

The first experiment includes a set of 11 features that were selected based on their frequent use by other related studies. For the second experiment, the same 11 features were included along with the features that had the highest correlation with the target attribute. We planned to include the top 50% of the features with the highest correlation with the target feature. Hence, the correlation of all 25 features was calculated as shown in Table 3 which also includes the feature description [17]. However, some of the highly correlated features were eliminated for the following reasons:

- Direct relationship with the target feature:

In each logline, the attributes with the highest significance were the action source, log action, and threat/content as shown in Table 4. These attributes showed direct relation to the produced recommended action and therefore could not be used for training. This observation was made when the J48 algorithm was applied; it showed an if/else relationship between these three features and the target feature as the model depended solely on these three attributes to produce an output. The same idea was noted with the rule attribute, it is dependent on the organization and has a straightforward relationship with the target attribute. If the rule was matched, its associated action is made. Thus, the mentioned features were not useful for training the model and instead caused underfitting.

- Target leakage:

The session end reason and flag features were also overlooked as they cause target leakage which occurs when a feature is a result of the target attribute, not the other way around. Hence, in a normal setting, these attributes would not be known until the action is already taken. This makes the mentioned features useless for our purpose of predicting the action to be taken.

Consequently, the previously mentioned features were not added to the second experiment, and therefore only two extra features, namely, category and application, presented high correlation and were added with an aim to improve the performance obtained in the first experiment. Features selected for the first and second experiments are summarized in Table 3.

**Table 3.** Set of features used in each experiment.

| Experiment 1: Features Commonly Used by Similar Studies | Experiment 2: Commonly Used Features Combined with the Features with the Highest Correlation |
|---|---|
| Source Port | Application |
| Destination Port | Category |
| NAT Source Port | Source Port |
| NAT Destination Port | Destination Port |
| Elapsed Time | NAT Source Port |
| Bytes | NAT Destination Port |
| Bytes Sent | Elapsed Time |
| Bytes Received | Bytes |
| Packets | Bytes Sent |
| Packets Sent | Bytes Received |
| Packets Received | Packets |
| | Packets Sent |
| | Packets Received |

**Table 4.** Dataset features description and their correlation.

| Feature | Description | Correlation |
|---|---|---|
| Action Source | Indicates whether the action taken was based on the application or policy. | 0.5 |
| Log Action | Specifies the log-forwarding profile that was applied to the session. | 0.49998 |
| Threat/content type | A subtype of traffic log: allow (allows the application), drop (drops the session before identifying the application), deny (drops the session after the application is identified and does not match any rule to allow it, or it matches a rule that blocks it). | 0.48175 |
| Session End Reason | The cause of the session's termination. | 0.44833 |
| Application | The application of the session, namely, HTML, DNS, Snapchat, WhatsApp . . . etc. | 0.35174 |
| Flag | Provides 32 bits of encoded information about the session. | 0.33422 |
| Rule | The name of the rule that was matched with the session. | 0.2996 |
| Category | The type of URL for the session. | 0.27856 |
| IP Protocol | The session's IP protocol. | 0.24317 |
| Source Zone (from) | The zone the session was sent from. | 0.18883 |
| Destination Zone (to) | The zone for which the session was intended. | 0.18882 |
| Destination Port | The destination port used by the session. | 0.18729 |
| Virtual System Name | Name of session's virtual system. | 0.17703 |
| Source Port | The port of the source used by the session. | 0.17622 |
| NAT Source Port | Network Address Translation source port. | 0.1461 |
| Virtual System | Virtual System mapped to the session. | 0.1292 |
| Repeat Count | The number of sessions with the same source IP, Destination IP, Application, and subtype seen within five seconds. | 0.06154 |
| Elapsed Time (sec) | Duration of the session. | 0.05686 |
| NAT Destination Port | Network Address Translation destination port. | 0.03336 |
| Bytes Received | The number of bytes received during the session. | 0.00984 |
| Bytes | The number of total bytes—sent and received—during the session. | 0.00918 |
| Packets Received | The number of packets received during the session. | 0.0076 |
| Packets | The number of total packets sent and received—during the session. | 0.00741 |
| Packets Sent | The number of packets sent during the session. | 0.00715 |
| Bytes Sent | The number of bytes sent during the session. | 0.00663 |
| Action (Target Class) | The action chosen to deal with the received session: allow, deny, drop (terminate silently), reset both (terminated, and send a TCP reset to both the sender and receiver). | |

### *3.4. ML and DL Algorithms*

The final step after feature selection is classification. In this step, the network traffic log's dataset is analyzed and the features are fed into the classifiers including ANN, NB, KNN, RF, and J48. Furthermore, we compared the performance of the classification phase of these algorithms in terms of accuracy, precision, recall, F-measure, and ROC values. This section lists and describes the ML and DL algorithms used in this study.

### 3.4.1. Artificial Neural Network (ANN)

ANN is one of the core techniques of DL that consists of three layers: input, hidden, and output layers. Each layer works on learning and assessing the given data and sending its output to the next layer to ultimately enable making predictions according to learning and analyzing the data. An ANN consists of a group of several perceptions or neurons that make up each layer. It is also known as a Feed-Forward Neural Network (FFNN), where inputs are processed only in a forward path. Furthermore, the output layer extracts information from preceding layers using the activation function to produce the final result. This activation function is used to transfer the input into the desired output through mathematical calculations, hence, it can be called the transfer function. Figure 3 illustrates a perceptron's architecture, which is essentially a general single layer ANN [18].

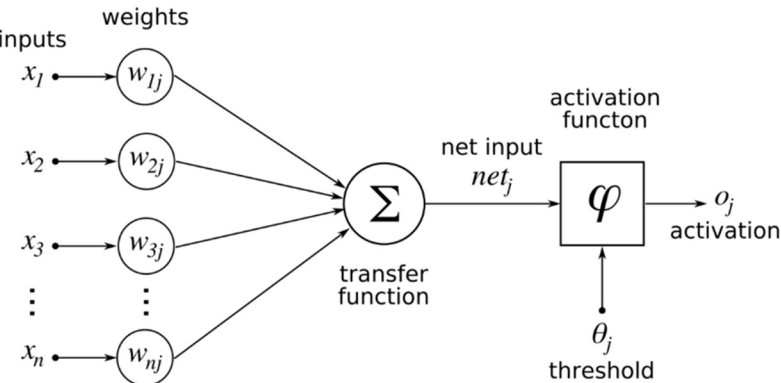

**Figure 3.** Perceptron architecture in its simplest form, showing the three main layers of an ANN.

Each activation function has a different equation. The general formula is shown in Equation (1), where x is the input, w is the weight, b is the bias weight value, and n is the number of input nodes starting from j = 1.

$$Y = \sum_{j}^{n}(w \times x) + b \tag{1}$$

Moreover, ANN algorithms are adaptive and scalable, which makes them suitable for dealing with large datasets and highly complex ML problems [9,11].

3.4.2. Decision Tree (DT)

DT is a supervised learning technique used for both regression and classification problem-solving. This algorithm is straightforward, using a tree representation with branch-like segments. Each internal node represents an attribute and branches the data into two distinctive groups. This is repeated until a class label is reached, which is represented by a leaf. Moreover, after fitting the tree, predictions can be made by tracing a path from the root to a leaf. In other words, in the beginning, the entire set of data is set at the root and based on different features observed in the data it is repeatedly split into two groups until reaching a set of class labels to make predictions. This algorithm can handle large datasets without mandating a complicated structure [8].

A major challenge when using the DT algorithm is the attribute selection method, as the selected attribute at each level plays an important role in splitting the data. There are two main measures used in attribute selection: information gain and Gini Index. As the training records are partitioned into smaller subsets, the entropy changes. This change in entropy is measured through the informational gain using Equation (2) where T represents the class labels and X represents a specific attribute.

$$\text{Gain}(T, X) = \text{Entropy}(T) - \text{Entropy}(T, X) = \sum_{i=1}^{J} P_i \log_2 P_i - \sum_{i=1}^{J} -P_r(i|a) \log_2 P_r(i|a) \tag{2}$$

The Gini Index measures how often a randomly chosen element is incorrectly labelled. Hence, attributes that have a lower Gini Index are preferred. The Gini Index can be calculated using Equation (3) by summing the probability of an item being chosen times the probability of a mistake in labelling that item [19].

$$\sum_{k \neq i} P_k = 1 - P_i \tag{3}$$

The j48 algorithm is an open-source optimized java implementation of the C4.5 DT algorithm [20]. It is an extension of the Iterative Dichotomiser 3 (ID3), as it is improved by including extra features to deal with high variance, missing values, etc. Considering

that it is essentially a DT, it calculates predictions based on the branching of the given data using the attribute values. In the same manner as DTs, the internal nodes represent the different features, the branches denote the different values given the feature, and the leaves represent the end results, the class labels [2,19].

### 3.4.3. K-Nearest Neighbors (KNN)

KNN is a nonparametric classification method based on supervised learning. It is one of the simplest and most straightforward algorithms and is widely used in practice. Moreover, it is used for both classification and regression. In both cases, once the training set is determined, the input consists of the k closest training samples in the dataset that are then used for prediction according to the category distribution among these k nearest neighbors [21]. However, the distribution among the sets may be uneven as some of them might have more examples than others. Hence, the algorithm's performance is significantly affected by the value chosen for the k parameter. The main calculations rely on distance formulas such as the Euclidean distance function, which can be calculated using Equation (4) used to calculate the distance between two points p and q.

$$d(p,q) = d(q,p) = \sqrt{(q_1 - p_1)^2 + (q_2 - p_2)^2 + \ldots + (q_n - p_n)^2} = \sqrt{\sum_{i=1}^{n}(q_i - p_i)^2} \quad (4)$$

In the KNN algorithm, the distance between the samples is repeatedly calculated throughout the training records and then the nearest K-number of observations in the training data is selected using the previously mentioned Euclidean distance. In simple words, the system finds the k nearest neighbors across the samples and uses the groups of the k nearest neighbors to weigh the cluster candidates. However, KNN has an issue when it comes to efficiency, as it requires time in assessing the test document against all the training set samples in order to assign it to one of the clusters [3,12].

### 3.4.4. Naïve Bayes (NB)

NB classifiers are a family of supervised learning algorithms. They are ML models based on a theory of probability that is used for classification tasks. The Bayes theorem in Equation (5) is the core that the classifier is based on.

$$P(A|B) = \frac{P(B|A)P(A)}{P(B)} \quad (5)$$

Using the Bayes theorem, we can estimate the possibility of A, the hypothesis, occurring, knowing that B, the evidence, has occurred. The theory here is that the features are conditionally independent, that is, the presence of one feature does not affect the others; hence, the name naïve [4]. Although this independence assumption is not usually correct in practice, the classifier nonetheless usually delivers competitive accuracy among other algorithms, along with its efficient computation and many other advantageous attributes that make NB commonly used in practice [4,5].

### 3.4.5. Random Forest (RF)

RF is a supervised learning ML technique used for solving classification and regression problems. It is considered an ensemble learning algorithm as it is based on combining the results of several models to improve its performance. As the name suggests, it combines multiple DTs each representing a subset of the datasets and averaging their prediction to improve the overall accuracy of the model. Consequently, when more trees are involved, it improves the model performance while preventing the issue of overfitting [15].

*3.5. Evaluation Metrics*

This section presents the evaluation metrics used for the performance assessment of the classification models. This paper evaluates the classification performance of the used models using classification accuracy, precision, recall, F-measure, and ROC.

The accuracy demonstrates the proportion of the total number of correct predictions, which is the value of successfully classified instances. The classification accuracy is calculated using Equation (6) by dividing the total number of predictions that were correct by the total number of predictions.

$$\text{Classification Accuracy} = \frac{\text{TP} + \text{TN}}{\text{TP} + \text{FN} + \text{FP} + \text{TN}} \tag{6}$$

Regarding the confusion matrix, it includes four variables which are True Positive (TP), True Negative (TN), False Positive (FP), and False Negative (FN). Table 5 illustrates the 2 × 2 confusion matrix [4]. Each row of the confusion matrix represents the instances in a predicted class and each column represents the instances in an actual class. Each variable indicates a different meaning used in assessing the performance of a model. For instance, TP implies that the presence of threats was correctly detected. On the other hand, TN suggests that the absence of threats was correctly predicted. FN indicates that the model erroneously predicted that there was no threat and FP shows that the model failed to detect the threat's presence [22].

**Table 5.** 2 × 2 confusion matrix.

|  |  | Predicted Label | |
| --- | --- | --- | --- |
|  |  | Anomalous | Normal |
| Actual Label | Anomalous | TP | FN |
|  | Normal | FP | TN |

Furthermore, these four variables are used to calculate the precision, recall, and F-measure. Precision calculated using Equation (7) quantifies the number of positives, "Anomalous" instances out of all the instances predicted as "Anomalous". As for recall, it also predicts the positive class predictions, however, as in Equation (8) it is calculated over the number of "Anomalous" instances in the dataset itself regardless of whether or not they were correctly predicted. Finally, the F-measure provides a single count that combines the values of the precision and recall in one number and is calculated using Equation (9) [23].

$$\text{Precision (P)} = \frac{\text{TP}}{\text{TP} + \text{FP}} \tag{7}$$

$$\text{Recall (R)} = \frac{\text{TP}}{\text{TP} + \text{FN}} \tag{8}$$

$$\text{F} - \text{Measure(F)} = 2 \times \frac{\text{P} \times \text{R}}{\text{P} + \text{R}} \tag{9}$$

ROC values include the ROC curve and ROC area. Fundamentally, the ROC curve is a plot that measures the sensitivity, the True Positive Rate (TPR) in the function of the FPR for different points of the parameter. Thus, the area under the curve is an indicator of how effective a parameter is at distinguishing between two groups [3].

**4. Experimental Setup**

To perform the experiments, multiple models were built using ML and DL algorithms. The models with ML algorithms were built using Weka 3.8.5 [20], and the DL model was built using Python 3.8 on the Google Collab notebook platform [21]. The number of instances used to perform the experiments was 112,532 instances, and a target class of four labels "Allow", "Drop", "Deny", or "Reset-Both" was used. For each experiment, a

different set of features were used. In the first experiment, 11 features were used which are: Destination port, Source port, NAT Destination port, NAT Source port, Bytes, Bytes Received, Bytes sent, Packets, Packets received, Packets sent, and Elapsed Time. In the second experiment, 13 features were used, which are the same as the previous list with the addition of Application and Category. In both experiments, the models were trained and built using 10-fold cross-validation.

Grid search with a Cross-Validation (CV) Parameter Selection algorithm was used in parameter tuning for all ML models. CV parameter selection works in the same way as K-fold cross-validation for training and testing a model by using different sets for evaluation, but here the purpose is to increase the accuracy, so it uses error rate as an evaluation matrix [22], while grid search uses all possible combinations of settings until reaching the optimal value.

The ANN model is built using two hidden layers of 20 neurons in each layer and a single output layer of 4 neurons. The Rectified Linear Unit (ReLU) activation function was used for the hidden layers, and the SoftMax function was used for the output layer. Table 6 shows the optimization values used for all the models applied in this study.

**Table 6.** Parameters optimization.

| Model | Parameter | Optimal Value |
|---|---|---|
| KNN | K-value | 1 |
| | Distance Function | Euclidean distance |
| RF | Number of Trees | 128 |
| | Max Depth | 16 |
| J48 | Max Depth | 120 |
| | Classification Criteria | Entropy |
| | Number of Hidden Layers | 2 |
| | Number of Neurons in Hidden Layers | 20 |
| ANN | Activation function in Hidden Layers | ReLU |
| | Number of Neurons in Output Layer | 4 |
| | Activation function in Output Layer | SoftMax |

## 5. Results and Discussion

The multiclass ML and DL models were evaluated in terms of accuracy, precision, recall, F-measure, and ROC in both experiments. The best values are demonstrated in Table 7 for experiments one and two.

**Table 7.** Experiment results.

| Model | Evaluation Matrix | Experiment 1 | Experiment 2 |
|---|---|---|---|
| | Precision | 0.989 | 0.993 |
| | Recall | 0.989 | 0.993 |
| KNN | F-measure | 0.989 | 0.993 |
| | ROC Area | 0.999 | 1 |
| | Accuracy | 98.93% | 99.33% |
| | Precision | 0.954 | 0.967 |
| | Recall | 0.954 | 0.967 |
| NB | F-measure | 0.954 | 0.967 |
| | ROC Area | 0.998 | 0.998 |
| | Accuracy | 95.43% | 96.69% |
| | Precision | 0.986 | 0.984 |
| | Recall | 0.986 | 0.984 |
| J48 | F-measure | 0.967 | 0.992 |
| | ROC Area | 0.996 | 0.999 |
| | Accuracy | 96.71% | 99.16% |
| | Precision | 0.991 | 0.996 |
| | Recall | 0.991 | 0.996 |
| RF | F-measure | 0.991 | 0.996 |
| | ROC Area | 1 | 1 |
| | Accuracy | 99.11% | 99.64% |
| | Precision | 0.779 | 0.929 |
| | Recall | 0.779 | 0.929 |
| ANN | F-measure | 0.779 | 0.929 |
| | ROC Area | 0.891 | 0.992 |
| | Accuracy | 77.87% | 92.92% |

As shown in Table 7, RF achieved the best performance for all evaluation matrices in both experiments with an accuracy of 99.11% and 99.64% for experiments one and two, respectively. In general, all models performed well in terms of all evaluation matrices. Moreover, as shown in Figure 4, all models achieved better results in the second experiment compared to the performance in the first experiment. This highlights the impact of including the application and category features, especially for the ANN algorithm as demonstrated in Figure 4d.

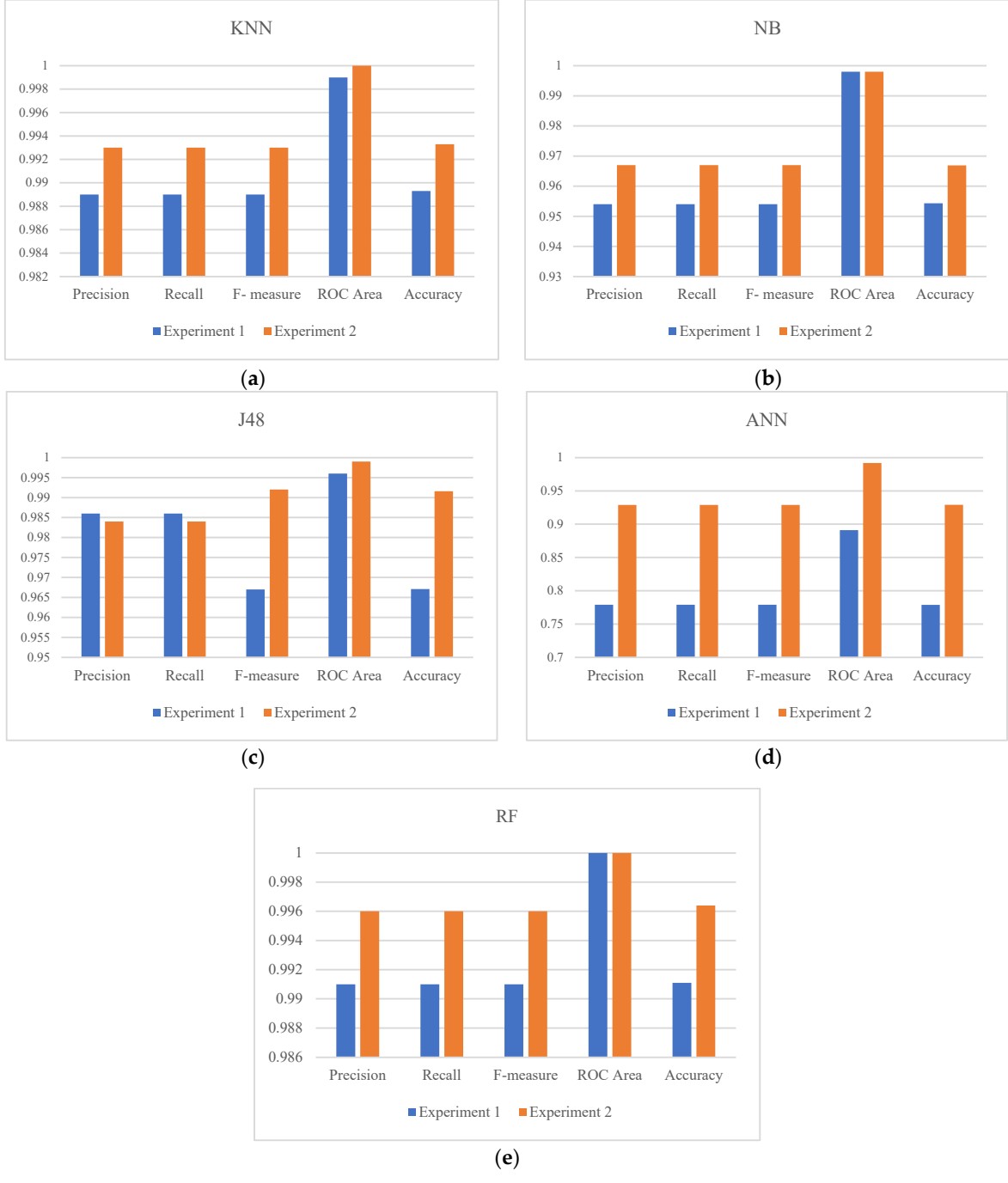

**Figure 4.** Comparison between results achieved in the first and the second experiment by each of the classifiers. (**a**): Performance analysis of the KNN classifier (**b**): Performance analysis of the NB classifier (**c**): Performance analysis of the J48 classifier (**d**): Performance analysis of the ANN classifier (**e**): Performance analysis of the RF classifier.

Hence, as all the models' performance improved in the second experiment, we further analyzed the models' performance by comparing the achieved accuracy in the second experiment. The results have shown that the RF classifier outperformed the other models and was closely followed by the KNN classifier as indicated in Figure 5.

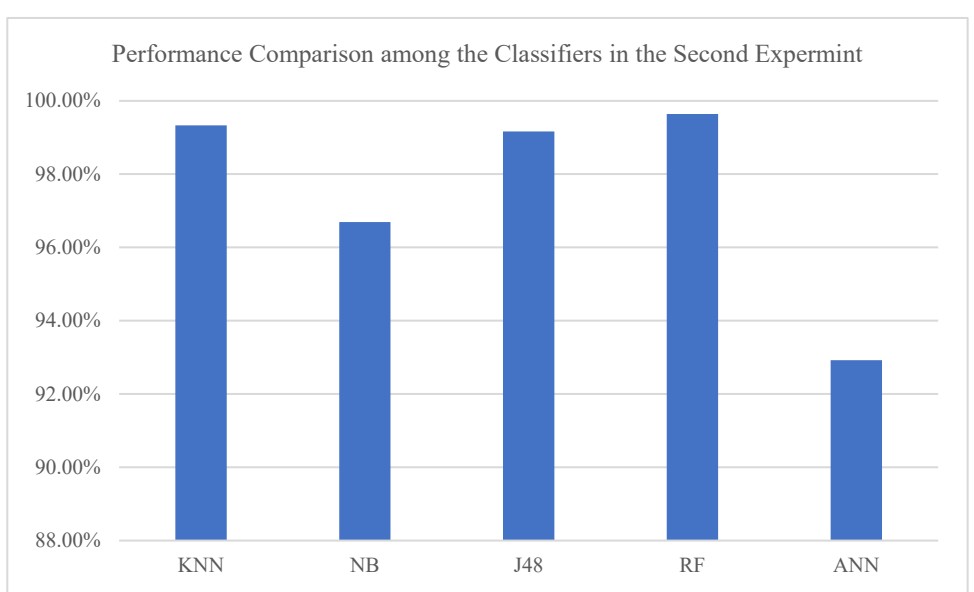

**Figure 5.** Comparison between the achieved accuracy for all classifiers in the second experiment.

To get a better evaluation, the confusion matrix was considered. Figures 6 and 7 show the confusion matrix when using the RF classifier, showing the number of correctly classified instances for both experiments.

```
=== Confusion Matrix ===

    a     b     c     d   <-- classified as
28131     2     0     0 |    a = allow
    0 27169     0   964 |    b = deny
    0     0 28133     0 |    c = drop
    0    35     0 28098 |    d = reset-both
```

**Figure 6.** Confusion matrix for RF algorithm in the first experiment.

```
=== Confusion Matrix ===

    a     b     c     d   <-- classified as
28132     0     0     1 |    a = allow
    0 27658     0   475 |    b = deny
    0     0 28133     0 |    c = drop
    1    10     0 28122 |    d = reset-both
```

**Figure 7.** Confusion matrix for RF algorithm in the second experiment.

The diagonal of the confusion matrix shows the correctly classified instances for each class. Each class contained 21,133 instances, and, as shown in Figure 6, more instances were classified correctly in the second experiment.

The difference between the first and second experiments was only the addition of two features in the second experiment, namely, application and category. Despite the difference

being minor, it resulted in better performance in all models. This shows the high impact of the application and the category of the website in the action that must be taken in handling it. Applying this to the real-world setup, we would indeed find that many websites are blocked based on their content and category; this supports the results achieved when analyzing both experiments. Moreover, the common factor between the two conducted experiments is that the RF algorithm achieved the best performance among all the models, which is believed to be due to RF being an ensemble algorithm which makes it a powerful algorithm. In addition, RF is one of the most commonly used algorithms in many fields as it obtains great performance while preventing overfitting. As RF trains multiple decision trees at the same time, each tree of them performs feature selection during the training process, and the power of all the trees is combined to make the final prediction. On the other hand, NB demonstrated low performance due to the fact that NB assumes that the dataset features are independent, and this assumption is in fact rarely true in most datasets. Moreover, NB works well with data that is high-dimensional and hence is widely applied to text classification problems. It's worth noting that the ANN model showed the lowest accuracy, which could be because it usually works with massive volumes of data with millions of records. Moreover, it is computationally expensive, and its complex structure makes it difficult for performance fine-tuning.

## 6. Conclusions

Firewalls are an important element of organizational network security, as they are the first line of defense in the network. Furthermore, firewalls can protect from external as well as internal attacks. Taking into account the importance of firewalls in system security, this study focused on building different ML and DL models that can classify the action that must be taken in response to sessions in firewall logs. A comparative analysis of five multiclass algorithms was performed to classify the action as "Allow", Drop", "Deny", or "Reset-both". A private dataset of 1,048,576 firewall logs was collected and used for training and evaluating the models. Furthermore, this study showed a comparison between the different features by conducting two experiments with two sets of features. A total of 11 features were included in the first experiment, and 13 features were included in the second experiment. The results of these experiments showed the impact of using the application and category of a website in selecting the proper action and how this improves the performance of a firewall. Moreover, all the models proposed in this study reached high accuracy, with the highest accuracy of 99.64% using the RF algorithm in the second experiment. This study supports the use of ML techniques in classifying the action of firewall logs automatically in a reliable and faster manner to improve the security of organizational networks. Moreover, the achieved results can contribute to improving the security and protection provided by firewalls and antivirus programs and building new techniques to prevent cyber threats.

**Author Contributions:** Conceptualization, M.A. (Malak Aljabri), A.A.A., R.M.A.M. and S.H.A.; methodology, M.A. (Malak Aljabri), R.M.A.M., S.H.A., M.A. (Menna Aboulnour) and D.M.A.; software, M.A. (Menna Aboulnour) and D.M.A.; validation, M.A. (Malak Aljabri), M.A. (Menna Aboulnour) and D.M.A.; formal analysis, M.A. (Malak Aljabri), M.A. (Menna Aboulnour) and D.M.A.; investigation, M.A. (Malak Aljabri), R.M.A.M., S.H.A., M.A. (Menna Aboulnour) and D.M.A.; resources, M.A. (Menna Aboulnour) and D.M.A.; data curation, S.H.A.; writing—original draft preparation, M.A. (Malak Aljabri), M.A. (Menna Aboulnour) and D.M.A.; writing—review and editing, M.A. (Malak Aljabri) and A.A.A.; visualization, M.A. (Menna Aboulnour) and D.M.A.; supervision, M.A. (Malak Aljabri); project administration, M.A. (Malak Aljabri) and A.A.A.; funding acquisition, M.A. (Malak Aljabri) and A.A.A. All authors have read and agreed to the published version of the manuscript.

**Funding:** The authors would like to thank SAUDI ARAMCO Cybersecurity Chair at Imam Abdulrahman Bin Faisal University for funding this project.

**Data Availability Statement:** Due to the confidentiality of the organization's information and data, the dataset used in this study are not publicly available and cannot be provided.

**Conflicts of Interest:** The authors declare no conflict of interest.

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
