# Peer review of "Classification of Firewall Log Data Using Multiclass Machine Learning Models"

_electronics, doi:10.3390/electronics11121851_

Round 1
Reviewer 1 Report
The aim of the paper is to evaluate and compare the performance of 6 ML and DL algorithms in analyzing network logs to determine the actions to be taken. 5 of them are commonly used ML algorithms. The sixth one - Artificial Neural Network (ANN) is rather a generic name for a core technique in DL, as it is mentioned in 3.4.1.
It is not clear the reason for mixing in the same experiment and comparing ML algorithms and ANN, as long as the authors are aware of the dependence of the performance of DL algorithms on the volume of input data and the available computing power.
The main reported contribution is the improved performance achieved in the second experiment by adding application and category features for websites. From this point of view, Figures 5 and 6 are not relevant in my view in supporting this conclusion and I would recommend the authors reconsider their opportunity.
There are many errors in citing references, which seem to be generated by the insertion of an additional [9] reference in the list without updating the reference numbers accordingly. Also, the reference Zhanpei et al. [14] in line 144 is not included in the list.
As a general rule, ”et al.” is used in citing a reference with 3 or more authors.
There are some acronyms that are not named clearly at their first occurrence in the text, e.g. LR (Linear Regression ?) and PSO (Particle Swarm Optimization ?).
In Figure 1, in the Machine Learning area, we should have RF instead of RB.
In line 248, I suppose Table 3 should be referred to rather than Table 2.
Please pay attention that Section 4 is called ”Results and Discussion” while Section 5 is ”Discussion”.
Reviewer 2 Report
There are two main problems with this paper. The first one is that it uses private data that cannot be checked or used to repeat the experiments. However, this is a minor problem. The main one is that the paper does not specify the final goal of the research and so we do not know why all these techniques have been applied to analyze the log. If the goal is to develop new rules for firewalls then the paper does not discuss this topic in detail. Furthermore, the paper selects a subset of available data and uses this subset to build the classification rules. However, the paper does not apply the classification rules it has computed to the whole dataset. So there is no experimental validation of the rules and no validation with respect to available data.
Reviewer 3 Report
This paper is weak and needs real and significant improvements.
In order to improve network secu- 28 rity, this study aims to build multiclass ML and DL models that can analyse firewall logs and clas- 29 sify the action to be taken in response to received sessions as “Allow”, “Drop”, “Deny”, or “Reset- 30 both”.
We suggest some points to the authors to tackle the main weaknesses risen in this paper.
We did not see clear contributions regarding this work as the main aim was just to use existing methods and no novel idea here
Also, the absrtact section show the idea of the current work but not in a sutabile and clear presenation, whcih need to be rewertien in a clear manner foucusign on the main contruction and the obtained results.
The introduction section present the general idea related to the prseneted topic but no foucusing also regarding the main aim of this paper.
The authors should connet the introduction details with the main aim of this reearch and how the main contrbution is needed beside the motivatoin
We look in to the main section in this paper , which the proposed work. We did not find a clear procedure and the given steps do not present a novel work
The results and the whole section of the results need more improvement and give more details regarding the results
Some new related works can be considered also...
Why the given equations have been colored with a red
Reviewer 4 Report
The title as well as the introduction raised expectations about the manuscript and research done. The topic you are addressing would be a relevant addition to existing literature. Thank you for this valuable contribution. I will structure my feedback in (a) general remarks (these comments cover feedback applicable in the entire manuscript), and (b) specific remarks (feedback on sentence and/or word level). The specific remarks can include a quote from your original manuscript to refer to a specific section. Some of the specific remarks will refer to specific lines (e.g., L 15/16).
General remarks
This is a well written and easy to read paper where authors tried to include an extra two features to existing work to improve the algorithms performance. I do acknowledge the potential of the manuscript and the topic. The Introduction and methodology in this paper are presented clearly. The research problem is well stated in the Introduction however the authors need to improve the abstract and the presentations of the results achieved. The problem statement is not clear in the abstract, what is the initial problem with the existing firewall logs? Highlighting the research contributions is vital. In the abstract you have to indicate: What is the problem? What did you do? What were your results? What did you learn (that is not already known in the literature) and what is the added value to theory? Please leave it to no more than 250 words.
The paper is well supported by mathematical equations but not pseudo codes. The tables’ font could be smaller which hopefully reduces the table size, for example Table 1 will reduce in size if the font is smaller. Results tables and figures were not presented well. For example, in Table 7 the data could be presented visually in a better way. Same with figure 5 and 6, the difference between the figures is hardly noticed. The overall manuscript still requires revisions to correct grammar mistakes, typo mistakes, make sure to present the full name before the acronym for example TWIDS. Define your acronyms before using them (use as few as possible). Finally, follow the journal template and check your complete reference list.
Specific remarks
The email for the corresponding author was repeated.
L 31/32 The first experiment was not mentioned
L 58 and L 93 have extra “ .”
L 65 There is an extra space
L 75 Missing space before ref [5]
L 134 Likewise, “L” capital letter
L 137 Full name required for the acronym “TWIDS”
L 146/147 how is the “Sparke-based log analyser” is different, elaborate more.
L 177 Not clear which the three papers you try to refer the readers to, elaborate more. I can see they were mentioned after.
L 205 you mean “the number of extracted features”
L 237 extra “.” in Figure 2
L 248 You mean Table 3?
L 278 It is not recommended to include citation in tables, e.g. in the title Table 3 you put [17]. Does that mean you copied all the table content from that reference? Or it is your own definition and correlation values.
Does the feature “Threat/content type” in Table 3 consider “both-reset” condition?
The “descriptions” in Table 3 should align with the “feature” in a better way. For example, “The cause of the session’s termination.” Is not aligned with “Session End Reason”
Figure 3 is blurry.
Figure 3 It is preferable not to include any figures which are borrowed from another paper, e.g. [18]
Equations are still under review tracker!
All equations parameters should be defined starting from eq. 2
L 359 “J.48” extra dot
L 360 citation for references and brief explanations are required for “C4.5 DT” and “Iterative Dichotomiser 3”.
L 372 “the classification”, T capital
Education 6 “TN’ ”or “TN”
L 394 Full name to “FPR”
L 479 typo mistake “Reset-bott”
L 412 section “4.2 parameters tuning” It is better to bring Table 4 forward to be included in this section before section 4.2.1
For reference list you should follow the journal template and check it for inconsistencies in spaces, commas, and year boldface, and italic font. E.g. check mistake in reference 9.
Round 2
Reviewer 2 Report
you have not addressed my observations.
Reviewer 3 Report
This paper has been revised by the authors in an excellent way. Therefor, they should highlight the main contribution in the proposed method section. Also, more results and comparisons can be presented.
